# Implications of variations in stream specific conductivity for estimating baseflow using chemical mass balance and calibrated hydrograph techniques

Ian Cartwright[1]

[1]School of Earth, Atmosphere and Environment, Monash University, Clayton, Vic. 3800, Australia

*Correspondence to*: Ian Cartwright (ian.cartwright@monash.edu)

**Abstract**

Baseflow to rivers comprises regional groundwater and lower salinity intermediate water stores such as interflow, soil water, and bank return flows. Chemical mass balance (CMB) calculations based on the specific conductivity (SC) of rivers potentially estimates the groundwater contribution to baseflow. This study discusses the application of the CMB approach in rivers from southeast Australia and assesses the feasibility of calibrating recursive digital filters (RDF) and sliding minima (SM) techniques based on streamflow data to estimate groundwater inflows. The common strategy of assigning the SC of groundwater inflows based on the highest annual river SC may not always be valid due to the persistent presence of lower salinity intermediate waters. Rather, using the river SC from low flow periods during drought years may be more realistic. If that is the case, the estimated groundwater inflows may be lower than expected, which has implications for assessing contaminant transport and the impacts of near-river groundwater extraction. Probably due to long-term variations in the proportion of groundwater in baseflow, the RDF and SM techniques cannot generally be calibrated using the CMB results to estimate annual baseflow proportions. Thus, it is not possible to extend the estimates of groundwater inflows using those methods, although in some catchments reasonable estimates of groundwater inflows can be made from annual streamflows. Short-term variations in the composition of baseflow also leads to baseflow estimates made using the CMB method being far more irregular than expected. This study illustrates that estimating baseflow, especially groundwater inflows, is not straightforward.

## 1. Introduction

Documenting the sources of water in rivers is required to understand catchment hydrology and to manage and protect water resources (Brunke and Gonser, 1997; Winter, 1999; Sophocleous, 2002; McCallum et al., 2010; Cranswick and Cook, 2015; Stoelzle et al., 2020). If rivers receive substantial groundwater inflows, the groundwater may be a source of contaminants (Bardsley et al., 2015; Crabit et al., 2016) and streamflow may be significantly reduced if near-river groundwater extraction occurs (Gleeson and Richter, 2018). These impacts potentially adversely affect the utility of surface water resources and riverine ecosystems. Understanding the water sources in rivers is also important for flood forecasting and assessing impacts of changes to climate or landuse. While it is well understood that rivers interact with several catchment water stores (e.g. groundwater, soil water, shallow perched riparian water, and water temporarily stored in the riverbanks: McCallum et al., 2010; Cranswick and Cook, 2015; Rhodes et al., 2017; Cartwright et al., 2018; Cartwright and Irvine, 2020) understanding those interactions is difficult.

Streamflow may be broadly divided into quickflow and baseflow (Hall, 1968; Nathan and McMahon, 1990; Tallaksen, 1995; Yu and Schwartz, 1999; Eckhardt, 2005). Quickflow (also sometimes referred to as storm flow or surface runoff) largely represents water derived from precipitation that contributes to streamflow soon after rainfall events. Baseflow represents water stored in the catchment that sustains streamflow between precipitation events. Regional groundwater may be a significant component of baseflow in gaining rivers; however, displaced soil water, interflow, bank return flows, snow melt, and/or water stored in floodplain pools can also be important (McCallum et al., 2010; Cranswick and Cook, 2015; Rhodes et al., 2017; Cartwright and Irvine, 2020; Stoelzle et al., 2020). The composition of baseflow may differ seasonally or between wet and dry years.

Long-term (years to decades), sub-daily to daily measurements of streamflow (hydrographs) are available for many rivers globally. Giver their ubiquity, numerous techniques (e.g., graphical separation based on minimum discharge, rainfall-runoff models, and recursive digital filters) have been used to estimate baseflow from streamflow (e.g., Nathan and McMahon, 1990; Gustard et al., 1992; Eckhardt, 2005; Brodie et al., 2007; Aksoy et al., 2008; Stoelzle et al., 2020). Groundwater and surface runoff generally have contrasting geochemistry, which allows baseflow to be estimated by geochemical mass balance. Most geochemical parameters (such as

major ions, stable and radioactive isotopes, dissolved gases, or nutrients) are not amenable to long-term

autonomous measurements, meaning that their use is largely confined to short-term studies such as separating

water sources over individual hydrograph peaks. Specific conductivity (SC) can, however, be readily measured

at similar frequencies and timescales to streamflow and many rivers have both long-term SC and streamflow

data (Yu and Schwartz, 1999; Gonzales et al., 2009; Sanford et al., 2011; Miller et al., 2014, 2015, 2016;

Cartwright et al., 2014; Rumsey et al., 2015; Hagedorn, 2020; Cartwright and Miller, 2021). While SC only

provides a general indication of water geochemistry, it is a valuable parameter to estimate baseflow

contributions, especially in catchments where groundwater is saline.

Baseflow calculations based on river hydrographs commonly group all delayed waters into the baseflow

component (Nathan and McMahon, 1990). By contrast, because the near-river intermediate water stores (e.g.,

interflow, bank storage water, and water stored in floodplain pools) are less saline than regional groundwater,

chemical mass balance (CMB) calculations are commonly assumed to reflect mainly groundwater inflows

(Gonzales et al., 2009, Miller et al., 2014, 2015, 2016, Rumsey et al., 2015; Hagedorn, 2020); although the input

of saline near-surface waters may also occur at the onset of high flows after prolonged dry periods. These

differences commonly result in CMB estimates of baseflow being lower than those based on the analysis of

river hydrographs (Cartwright et al., 2014; Lott and Stewart, 2016; Rammal et al., 2018; Saraiva Okello et al.,

2018; Chen and Teegavarapu, 2019; Hagedorn, 2020). Some studies (e.g. Cartwright et al., 2014) used those

differences to partition streamflow into groundwater inflows, intermediate stores, and surface runoff. Other

studies (e.g., Stewart et al., 2007; Gonzales et al., 2009; Zhang et al., 2013; Rammal et al., 2018; Saraiva Okello

et al., 2018) have used the CMB to parameterise the recursive digital filters (RDF) and sliding minima (SM)

techniques in order to use those to estimate groundwater inflows. Given that the different techniques have

different errors and assumptions, this may be problematic (Yang et al., 2021). However, from a pragmatic

viewpoint of extending estimates of groundwater inflows to time periods where there is no SC data or to similar

catchments, such parameterisation efforts are valuable.

### 1.1. Conceptualisation of baseflow

Baseflow is commonly conceived of as varying as shown in Figure 1. Total baseflow inflows are lower than surface runoff and the deeper catchment stores that contribute to baseflow, such as groundwater, are assumed to have longer wavelength and lower amplitude variations than the shallower intermediate stores. This results in groundwater inflows being the dominant source of streamflow only during low summer flows, whereas the streams may be sustained by baseflow from the combined catchment stores during low flow periods throughout the year (Fig. 1).

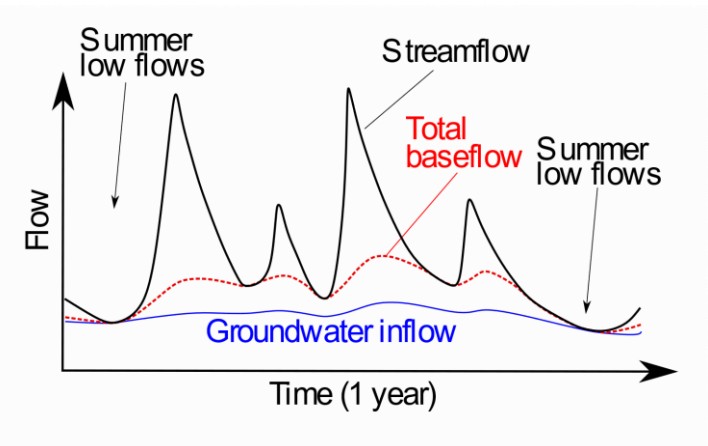

**Fig. 1.** Conceptualisation of baseflow. Over the water year, baseflow may dominate streamflow during successive low flow periods; however, groundwater inflows may be dominant only during low summer flows. Both the groundwater inflows and total baseflow are conceived to vary smoothly.

Baseflow separation techniques based on streamflow data are based on this conceptualisation. Recursive digital filters separate smooth longer wavelength baseflow inputs from shorter wavelength quickflow (Nathan and McMahon, 1990; Chapman, 1999; Eckhardt, 2005). Likewise, graphical baseflow separation techniques consider that baseflow varies in a regular manner between periods of low discharge (Gustard et al., 1992; Aksoy et al., 2008; Stoelzle et al., 2020). The variation of baseflow in Fig. 1 is consistent with our broader understanding of hydrogeology. Recharge generally occurs at slower rates than runoff and groundwater elevations change more slowly than river levels. The timescales that floodplain pools or riverbanks fill and drain are shorter than those over which groundwater responds but again are longer than the streamflow response to rainfall.

Hydrograph separations carried out over individual discrete flow events using major ions, stable isotopes, and/or radioisotopes (Sklash and Farvolden, 1979; Uhlenbrook et al., 2002; Klaus and McDonnell, 2013; Tweed et al.,

2016) also commonly conclude that baseflow varies smoothly over flow events. Many of those detailed studies are from relatively small headwater catchments and it is less clear whether baseflow in larger rivers varies in a similar regular manner. Larger rivers aggregate water from numerous subcatchments, each of which may have a different contribution of baseflow at any given time. For example, bank storage waters may continue to contribute to some parts of catchments for many months but cease relatively quickly in other areas (McCallum

et al., 2010; Cartwright and Irvine, 2020). Additionally, the discharge rate of groundwater depends on the hydraulic gradient between the groundwater and the river which is likely to vary both temporally and spatially.

### 1.2. Objectives

This paper discusses some of the issues involved in estimating baseflow using streamflow and SC data. Firstly, it assesses how frequently rivers are fed predominantly by groundwater. Resolving this question is important

for carrying out CMB calculations, especially understanding the length of the SC record that is required. Secondly, it discusses whether CMB technique can be used to calibrate streamflow-based techniques to provide similar annual baseflow estimates. If this is feasible, it would permit groundwater inflows to be estimated at times or locations where SC data is not available. Finally, it assesses if groundwater inflows vary smoothly over both short (weeks to months) and longer (years to decades) timeframes. These questions are addressed using

long-term streamflow and SC data from several perennial and intermittent rivers in Victoria (southeast Australia). The results of this study are important for a broader understanding of how the quantity and composition of baseflow varies over time.

### 2. Catchments

The study areas are the Barwon, Corangamite, Goulburn and Loddon catchments from southeast Australia.

These are described below with additional details presented in the Supplement. The Barwon catchment has an area of ~2700 km$^2$ and mainly consists of gently undulating plains comprising Piocene-Pleistocene Newer Volcanics Province basalts interbedded with marine and freshwater sediments (Dahlhaus et al., 2008;

Cartwright et al., 2013, 2014). The plains are mainly used for dryland grazing and crops. There are minor areas of tree plantations and remnant native forest on the hills at the catchment margins where basement Mesozoic to Cainozoic sediments crop out (Fig. S1). Annual rainfall increases southward from 600 to 1050 mm with July to September (the austral winter) the wettest months (Bureau of Meteorology, 2021). Total annual streamflows in the Barwon River increase downstream and 60% of the annual flow occurs between July and September (Department of Land, Environment, Water and Planning, 2021). Shallow groundwater in the Barwon catchment typically has SC values of 5000-25,000 $\mu$S cm$^{-1}$ (Dahlhaus et al., 2008; Cartwright et al., 2013, 2014) and SC values in the Barwon River are up to 25,700 $\mu$S cm$^{-1}$ (Department of Land, Environment, Water and Planning, 2021).

The Corangamite catchment is also located mainly on the basalt plains of the Newer Volcanics Province (Department of Jobs, Precincts and Regions, 2021: Fig. S2). The catchment includes several permanent and intermittent saline to hypersaline lakes that represent groundwater discharge points in topographic lows in the basalt surface. Annual rainfall is 720 mm and, as for the Barwon catchment, rainfall and streamflow are higher in winter (Bureau of Meteorology, 2021; Department of Land, Environment, Water and Planning, 2021). The catchment has also been largely cleared for dryland agriculture (Fig. S2). Groundwater SC values vary from ~1000 $\mu$S cm$^{-1}$ in the south of the catchment to >50,000 $\mu$S cm$^{-1}$ around the saline lakes. Groundwater is only locally used (mainly for stock watering). The streams in this catchment vary from intermittent to perennial and have SC values of up to 30,000 $\mu$S cm$^{-1}$ (Department of Land, Environment, Water and Planning, 2021).

The Goulburn catchment (Fig. S3) extends from the low-relief Riverine Plain of the Murray Basin southwards to the highlands of the Victorian Alps (Lawrence, 1988; Cartwright and Weaver, 2004). The shallowest sediments on the Riverine Plain are the heterogeneous terrestrial sands, silts, and clays of the Pliocene to Holocene Shepparton Formation and the contiguous Quaternary Coonambidgal Formation. These sediments onlap basement Proterozoic granites and metamorphosed turbidites in the highlands. Annual rainfall decreases northwards from 675 to 450 mm and rainfall and streamflow are again higher in the winter months (Bureau of Meteorology, 2021; Department of Land, Environment, Water and Planning, 2021). Flows in the middle and lower Goulburn River are controlled by water release from the Lake Eildon reservoir (3.3x10$^9$ m$^3$ capacity) but

many of the tributaries are unregulated and vary from perennial to intermittent. The middle and lower parts of the catchment have been largely cleared and comprises a mix of dryland and irrigated agriculture. The highlands include remnant native eucalyptus and plantation forests. Groundwater SC is mainly <5000 μS cm$^{-1}$ although local zones of higher salinity (SC values up to 20,000 μS cm$^{-1}$) groundwater exist in the east of the catchment (Cartwright and Weaver, 2004); river SC values are generally <5000 μS cm$^{-1}$ (Department of Land, Environment, Water and Planning, 2021).

The Loddon catchment (Fig. S4) is also part of the Riverine Plain of the Murray Basin and has similar geology and hydrogeology to the Goulburn catchment (Lawrence, 1988). Except for remnant native forest on basement rocks at the margins of the catchment, the catchment is largely cleared and predominantly used for dryland agriculture. Annual rainfall decreases northwards from 580 to 420 mm, and rainfall and streamflow are higher in the winter months (Bureau of Meteorology, 2021; Department of Land, Environment, Water and Planning, 2021). The streams in this catchment vary from intermittent to near-perennial and have SC values up to 40,000 μS cm$^{-1}$. Shallow groundwater in the Loddon catchment typically has SC values of 7000-30,000 μS cm$^{-1}$ (Department of Land, Environment, Water and Planning, 2021).

## 3. Materials and Methods

### 3.1. Data sources

The study uses data from gauging stations in the Barwon, Corangamite, Loddon and Goulburn catchments (Table 1, Supplement). The catchments upstream of the gauging stations have areas ranging from 5 to 2850 km$^2$ and are not impacted by major water storages or by near-river groundwater extraction. All of the studied catchments are from the low-relief parts of the catchments that have largely been cleared of native vegetation (Figs S1-S4). Daily mean streamflow (m$^3$ sec$^{-1}$) and mean SC (electrical conductivity referenced to 25 $^o$C in μS cm$^{-1}$) are from the Department of Environment, Land, Water and Planning (2021). These two parameters are measured at identical locations and times, and the daily values represent the arithmetic means of measurements made at intervals of 15 minutes to four hours. The length of time over which continuous SC data have been measured varies; however, many stations have records between 1994 and 2020 and that period was adopted in this study. Streams in southeast Victoria record low flows over the austral summer to autumn (typically January

to April). For this study the water year was defined as commencing in September, which is when the peak flows occur. Previous studies in southeast Australia (Cartwright et al., 2014; Cartwright and Miller, 2021) used July as the start of the water year; however, some intermittent streams in the Loddon catchment have relatively high SC values in June and July. If the July date was retained, the calculated SC of baseflow in successive water years would occasionally be from the same flow period. Rainfall data are from the Bureau of Meteorology (2021). Correlations are designated as being good with $R^2$ values $\geq 0.7$, moderate where $0.5 \geq R^2 < 0.7$, and poor where $R^2 < 0.5$.

### 3.2. Chemical mass balance

Baseflow (assumed to be mainly saline groundwater inflows) on day i ($b_i$ in $m^3 \, sec^{-1}$) was calculated from:

$$b_i = q_i \frac{SC_r - SC_s}{SC_b - SC_s} \qquad (1)$$

(Yu and Schwartz, 1999), where $SC_r$, $SC_b$, and $SC_s$ are the SC values of the river, baseflow, and surface runoff, respectively. This method requires that: 1) $SC_s$ and $SC_b$ are well defined; 2) SC behaves conservatively; and 3) there is a large contrast between $SC_r$ and $SC_b$. As noted above, the salinity of groundwater in these catchments is generally several orders of magnitude higher than that of rainfall. Reactive species (e.g. $NO_3^-$ or $HCO_3^-$) do contribute to SC; however, there is a robust correlation ($R^2$ values typically $>0.95$) between SC values and the concentration of conservative ions such as Cl in both groundwater and surface water (Cartwright et al., 2013, 2014, 2017). Initially a value of $SC_s$ of 50 $\mu S \, cm^{-1}$ was adopted, which corresponds to the lowest $SC_r$ recorded at high streamflows in these catchments over the study period. In common with other studies (Sanford et al., 2011; Miller et al., 2014, 2015, 2016; Cartwright et al., 2014; Rumsey et al., 2015), $SC_b$ is estimated from the SC of the river during low flows. Two methods for estimating $SC_b$ were used. The *Variable SC* approach estimates daily $SC_b$ values by interpolating between high $SC_r$ values in successive water years , which assumes that the river is entirely fed by groundwater each year during low flows (as in Fig. 1). Adopting the highest $SC_r$ value as $SC_b$ risks outlier values caused by the flushing of solutes after prolonged dry spells being used (Miller et al., 2014), which results in $b_i$ being underestimated. To overcome that potential problem, $SC_b$ was assigned as the 99[th] percentile of $SC_r$ as suggested by Miller et al. (2014, 2016) and Rumsey et al. (2015). The *Constant*

*SC* approach uses the highest $SC_b$ value from the variable SC calculations . This may be a valid assumption if

the SC of groundwater is relatively constant over time and the rivers are entirely fed by groundwater only during

low flow periods in drought years (with bank return waters and/or interflow providing some input at other

times). The use of 99[th] percentiles and interpolation results in a small number of days where $b_i$ calculated using

Eq. (1) exceeds $q_i$; for these, $b_i$ was set to $q_i$.

Baseflow and the baseflow index (BFI $= \sum b_i / \sum q_i$) for the individual water years and the entire records were

calculated from the daily values. Periods of zero flow are distinguishable from missing data and these are

included ($q_i = 0$, $b_i = 0$ on those days). Some of the gauges record high $SC_r$ values in zero flow periods,

presumably from evaporated stagnant water at the gauge. These $SC_r$ values are not considered in the calculation

of $SC_b$. The streamflow records for most sites are near complete (>99%) and small (<5 day) gaps in streamflow

were estimated by linear interpolation. As with streamflow, short gaps (<5 days) in $SC_r$ were infilled by linear

interpolation. For longer gaps, the annual baseflow ($b_a$) was adjusted for the actual number of days of data (i.e.

in a year with SC data on 95% of the days $b_a = \sum b_i / 0.95$). To avoid excessive errors from missing data, only

results for years where both SC and streamflow data (including zero flows) were recorded on >90% of the days

prior to infilling of any gaps are reported.

**3.3.  Estimates of baseflow based on streamflow**

Baseflow separation based on the Institute of Hydrology sliding minimum streamflow (SM) method as

implemented in the USGS Groundwater toolbox (Barlow et al., 2017) was used with the streamflow data.

Comprehensive descriptions of this method are provided by Gustard et al. (1992), Hisdal et al. (2004), Brodie

et al. (2007), Aksoy et al. (2008), and Barlow et al. (2017). This method identifies significant streamflow minima

that are plausibly where baseflow constitutes the total flow and calculates daily baseflow by interpolating

between those minima. The factor (f) used to identify the minima or turning points in streamflow was set at 0.9

(which is the value generally used with this method). The minimum discharge is assessed over non-overlapping

time periods (commonly referred to as the block size, N). Increasing N increases the spacing of the periods that

the stream is assumed to be only fed by baseflow, which decreases the calculated BFI (Stewart et al., 2007;

Stoelzle et al., 2020). N is commonly estimated from catchment area (Aksoy et al., 2008); however it is difficult

to verify that approach. An alternative approach is to use breakpoints in N vs. BFI trends to define N values for different baseflow components (Stoelzle et al., 2020). Here the approach of Stewart et al. (2007), which estimates N by bringing the BFI from this method into agreement with the BFI from the CMB method, is used. N is an integer and the value that produced the closest agreement was adopted. The BFI is significantly more sensitive to the value of N than f (Aksoy et al., 2008) and variations in f were not considered. As above only results for years where flow data was recorded on >90% of days prior to infilling are reported

The recursive digital filter:

$$b_i = \frac{(1-BFI_{max})ab_{i-1}+(1-a)BFI_{max}}{1-aBFI_{max}} q_i \qquad (2)$$

(Eckhardt, 2005) was also used to estimate baseflow. In Eq. (2), a is the recession constant which was estimated from the falling limbs of the hydrograph. following Nathan and McMahon (1990) and Eckhardt (2005). a varies between 0.92 and 0.95 with a median value of 0.93 (Table S1). $BFI_{max}$ is the maximum value of the baseflow index that can be calculated using the filter. While $BFI_{max}$ is commonly assigned using catchment characteristics (Eckhardt, 2005), here the value of $BFI_{max}$ that produces the same BFI as the CMB is used, which is similar to the approach of Gonzales et al. (2009) and Saraiva Okello et al. (2018). The filter was applied in a single pass with the condition that $b_i \leq q_i$. Again, only results for years where flow data was recorded on >90% of the days are reported.

## 4. Results

### 4.1. Variation in streamflow and specific conductivity

Figure 2 shows the temporal variation of streamflow and SC records for one site in each catchment; the records for the other sites are similar. The main Barwon River sites are near perennial (<5% of zero streamflow days); however, some of the Barwon tributaries have more significant cease-to-flow periods with up to 59% of days of zero streamflow (Table 1).

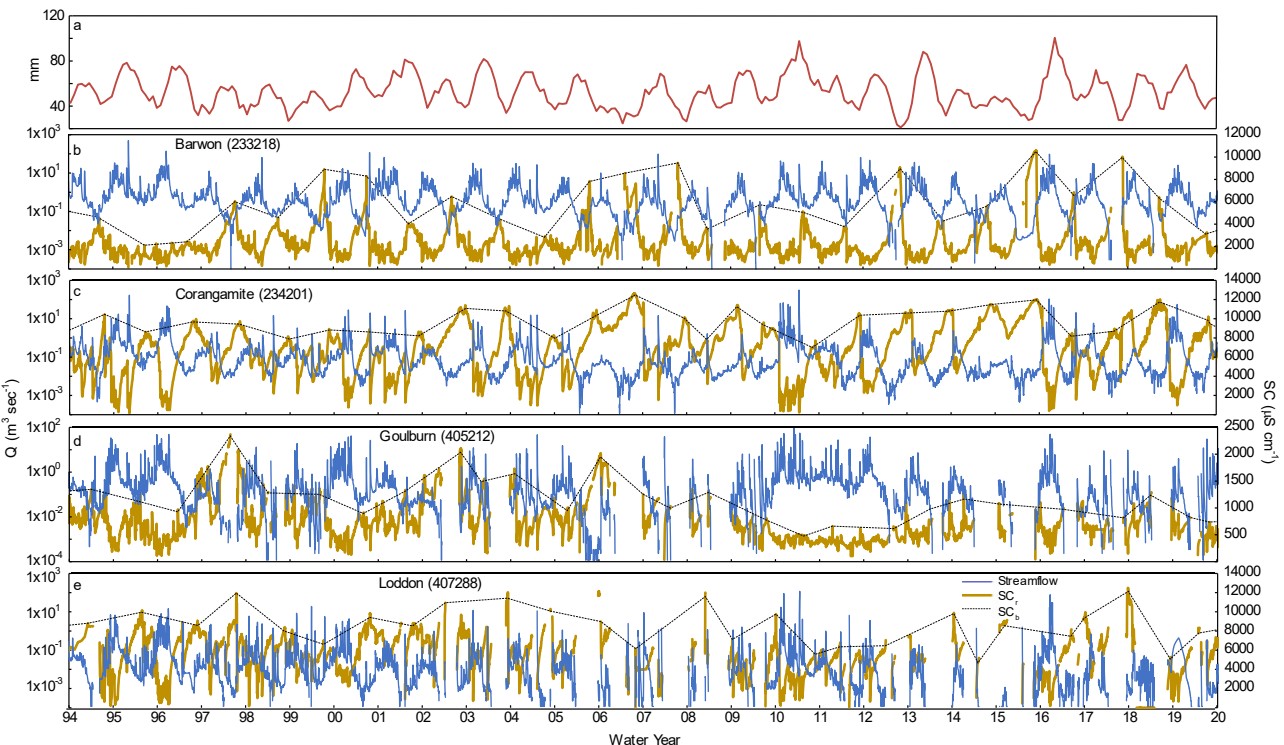

**Fig. 2a.** Variations in rainfall (monthly running mean) in southeast Australia (data from Barwon Catchment, station 90008: Bureau of Meteorology, 2021). **2b-2e**. Variation in streamflow (Q) and river SC ($SC_r$) for one station in the Barwon, Corangamite, Goulburn, and Loddon catchments (data from Department of Environment, Land, Water and Planning, 2021). $SC_b$ is the SC of baseflow calculated by interpolation between annual maximum $SC_r$ values.

The overall percentage of zero streamflow days in the other catchments are Corangamite 0-17%, Goulburn 64-33%, and Loddon 1-46%. $SC_r$ values are highest in the late summers when streamflow is lowest. However, the maximum $SC_r$ commonly varies between years. $SC_r$ values are generally higher in years of low rainfall when total streamflows are lowest. This variability is also evident in the $SC_r$ variations with streamflow (Fig 3). While there are broad inverse correlations between $SC_r$ and streamflow, there is a wide range of $SC_r$ values at the lower streamflows that reflects the year-on-year variability in stream salinity.

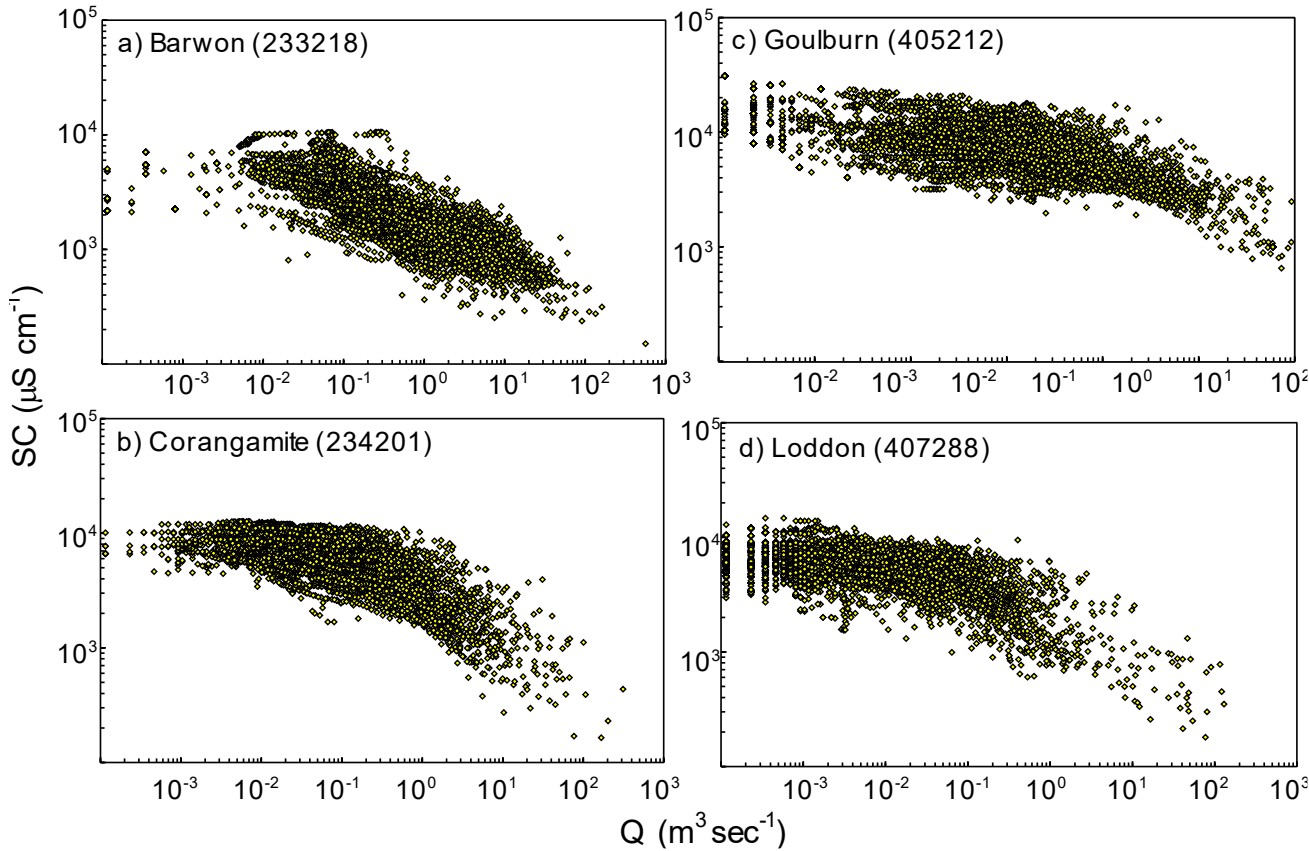

**Fig. 3.** Variation in streamflow (Q) and river SC for one station in the Barwon (**3a**), Corangamite (**3b**), Goulburn (**3c**), and Loddon (**3d**) catchments (data from Department of Environment, Land, Water and Planning, 2021).

### 4.2. Chemical mass balance estimates of baseflow

Values of BFI calculated using Eq. (1) and the two strategies for estimating $SC_b$ are shown in Figs 4 and S5-S8 and Table 1. Total BFI estimates calculated using the variable $SC_b$ approach are higher (0.13-0.50) than those calculated using the constant $SC_b$ value (0.04-0.37) and the annual BFI values are also higher. There is no *a priori* reason for assuming that rivers, even in relatively dry climates, are sustained wholly by groundwater inflows during low flows every year. Previous geochemical studies indicate that bank return flows are important in the Barwon catchment (Cartwright et al., 2014; Howcroft et al., 2019) and they are likely to occur in most rivers (McCallum et al., 2010; Cranswick and Cook, 2015; Rhodes et al., 2017). Using a constant $SC_b$ produces more regular yearly BFI vs. yearly streamflow trends (Figs 4, S5-S8; Table 1). Additionally, the constant $SC_b$ value results in a better correlation of total and yearly BFI values from different sites in the same catchments. Sites on the same river (such as those in the Barwon catchment) would be expected to have similar yearly BFI values or total BFI values that showed a coherent trend (e.g. systematically increasing or decreasing

downstream). This is the case where a constant $SC_b$ value is adopted but not where $SC_b$ varies annually (Table 2).

There are several other uncertainties in these calculations. Given the high salinity of the baseflow in these catchments, BFI is relatively insensitive to $SC_s$. Allowing $SC_s$ to vary between 25 and 75 μS cm$^{-1}$ results in an uncertainty in total and annual BFI values of <5% and does not change the correlations. Uncertainties in $SC_r$
values resulting from logger errors is probably less than that of streamflow. For example, most high-quality commercial SC loggers quote a precision of <1%, whereas the precision of streamflow in southeast Australia largely resulting from errors in the rating curves was estimated as typically ±5% (McMahon and Peel, 2019). These uncertainties are difficult to assess but they do not significantly impact the BFI estimates.

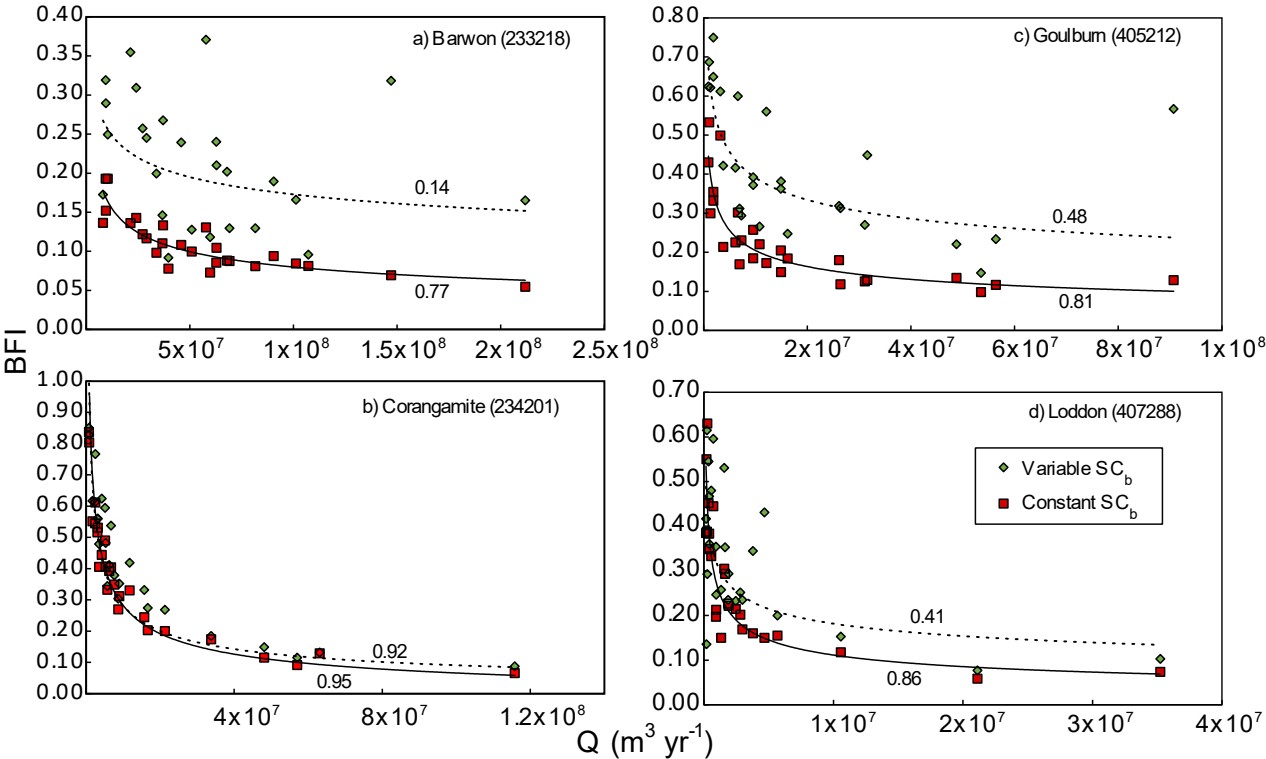

**Fig. 4.** Variation in streamflow (Q) and annual BFI calculated from CMB using the variable and constant SC calculations of $SC_b$ for one station in the Barwon (**4a**), Corangamite (**4b**), Goulburn (**4c**), and Loddon (**4d**) catchments. Lines are power law fits with $R^2$ values indicated.

### 4.3. Baseflow estimates from streamflow variations

As noted above, estimates of baseflow made using techniques based on streamflow data using default parameters would be higher than those resulting from CMB, largely due to the presence of low SC intermittent waters in the baseflow component. Adjusting the SM technique by varying N in Eq. (2) and the RDF by varying $BFI_{max}$ in Eq. (3) allows the total BFI estimates from these methods to be brought into agreement with those calculated using the CMB (Tables 1 and S1). For the SM technique, N values are as high as 35 days and are higher in catchments with low BFI values (Table S1). Values of $BFI_{max}$ are inversely proportional to the calculated BFI, and are as low as 0.07 (Table S1). Despite the long-term BFI values being brought into agreement, the annual BFI values estimated using these three techniques are commonly poorly correlated (Figs 5, S9-S12; Table 1). This lack of correlation is apparent regardless of which strategy is adopted to calculate $SC_b$. There is little difference in the degree of disagreement at high or low BFI values (Figs 5, S9-S12), which precludes systematic differences in behaviour between low streamflow years that have generally higher BFI values and high streamflow years with lower BFI values. There are also no significant differences in the correlation of BFI values and area, percentage flow, or the range of $SC_r$ values (which precludes the possibility that groundwater inflows may be easier to calculate in more saline rivers).

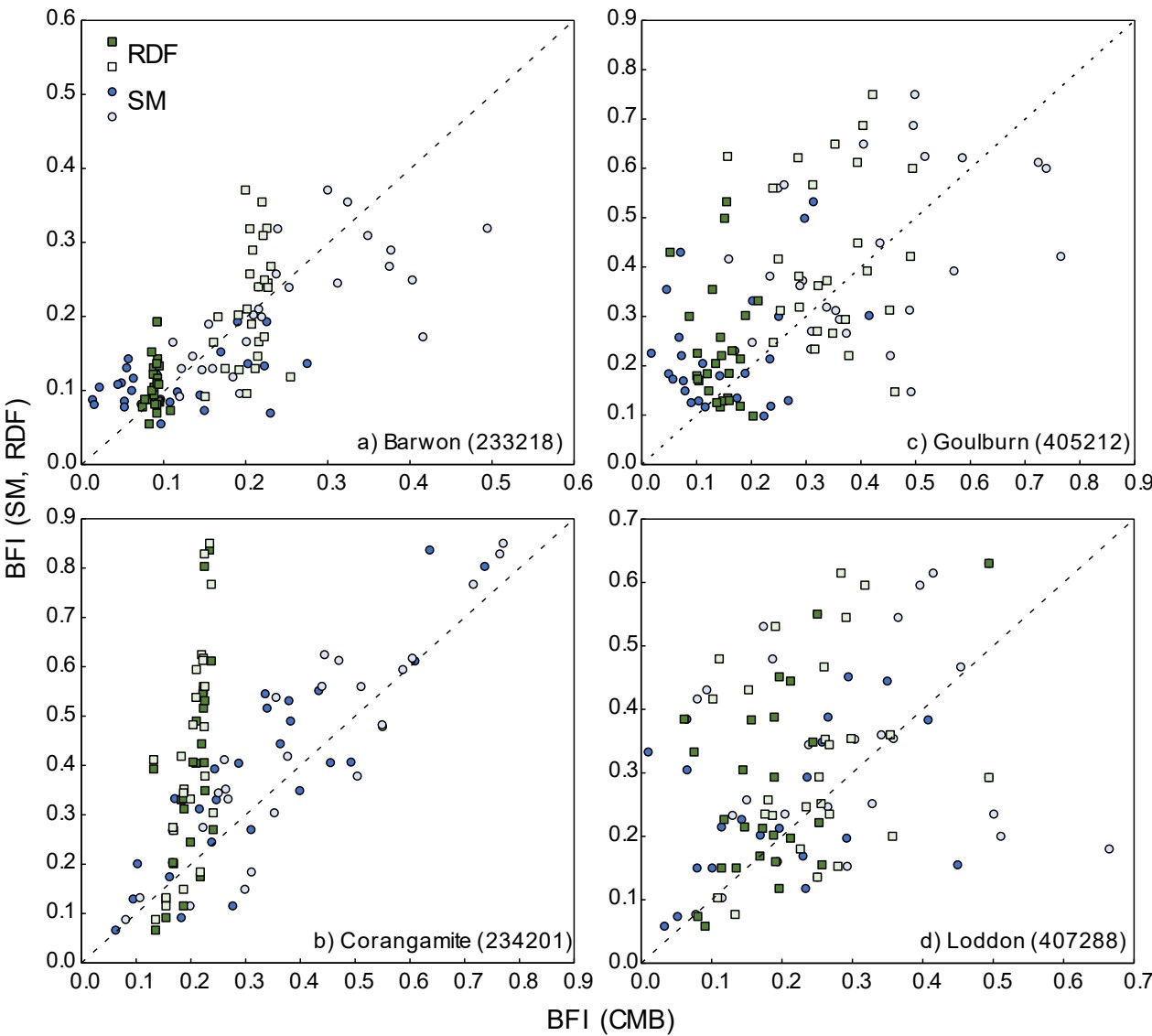

**Fig. 5.** Comparison of annual BFI estimates calculated from the RDF and SM methods with that from the CMB using the variable (open symbols) and constant (closed symbols) $SC_b$ approach for one station in the Barwon (**5a**), Corangamite (**5b**), Goulburn (**5c**), and Loddon (**5d**) catchments. Dashed lines are the 1:1 relationship.

## 5. Discussion

The Victorian rivers used in this study have long and near complete records of SC and streamflow records and occur in regions where groundwater is commonly saline. This makes them ideal for assessing the CMB technique and comparing this with other methods of estimating baseflow.

### 5.1. Quantifying baseflow using chemical mass balance

The $SC_r$ of these rivers are higher during low flow periods in low rainfall years compared with low flow periods in years of higher rainfall (Fig. 2). Rivers elsewhere show similar inverse correlations between annual SC values and rainfall (Hagedorn, 2020). The net SC of groundwater inflows may vary over time if groundwater from different parts of the catchment contributes differently to streamflow. However, given that bank storage waters may contribute to streamflow for months or years and may only drain after several years of low flows (McCallum et al., 2010; Cranswick and Cook, 2015; Cartwright and Irvine, 2020), it is more likely that regional groundwater only dominates baseflow during prolonged (multi-year) drier periods. If that is the case, the BFI estimates made using the variable $SC_b$ approach may overestimate the groundwater contribution (Fig. 4, Table 1). Additionally, where $SC_r$ is well correlated with streamflow, sporadically-measured SC values could be used to estimate groundwater inflows from the streamflow records (Miller et al., 2015). However, in the catchments discussed here, the year-on-year differences in stream salinity produces a broad range of $SC_r$ values at any given streamflow (Fig. 3) which precludes that approach. The constant $SC_b$ approach assumes that the SC groundwater is unvarying, which may also be an oversimplification. Adopting a hybrid approach that interpolated between the $SC_r$ values in low flow years would yields BFI estimates between the two sets of calculations. While this may be a more realistic model, it involves a significant amount of additional judgement in assigning $SC_b$ values.

### 5.2. Short-term variations in baseflow

The RDF and SM techniques produce smoothly varying baseflow inputs (Fig. 6) that conform to our perceptions of how baseflow should vary (Fig. 1). This is the case both where these methods are used with their default settings or where they have been modified to extract estimates of long-term baseflow or groundwater inputs (as in this study and Stoelzle et al., 2020). By contrast, $SC_r$ values in southeast Australian streams vary considerably over days to weeks (Cartwright and Miller, 2021), and this results in an irregular baseflow signal from the CMB method (Fig. 6). Similar irregular $SC_r$ variations and baseflow signals occur elsewhere (Hagedorn, 2020; Yang et al., 2021) and most probably reflect variations in the baseflow components. The contribution of bank return waters, waters draining from the floodplain, and interflow will vary in importance in different parts of the catchment over different timescales. Changes to the relative elevations of the water table and the rivers are likely

to also be spatially variable. This variability in catchment processes potentially leads to the composition of baseflow being highly variable over short timescales. Whether the total baseflow input is relatively smooth and the variation in $SC_r$ reflects the composition of baseflow or whether there are short-term variations in total baseflow is not constrained by these data.

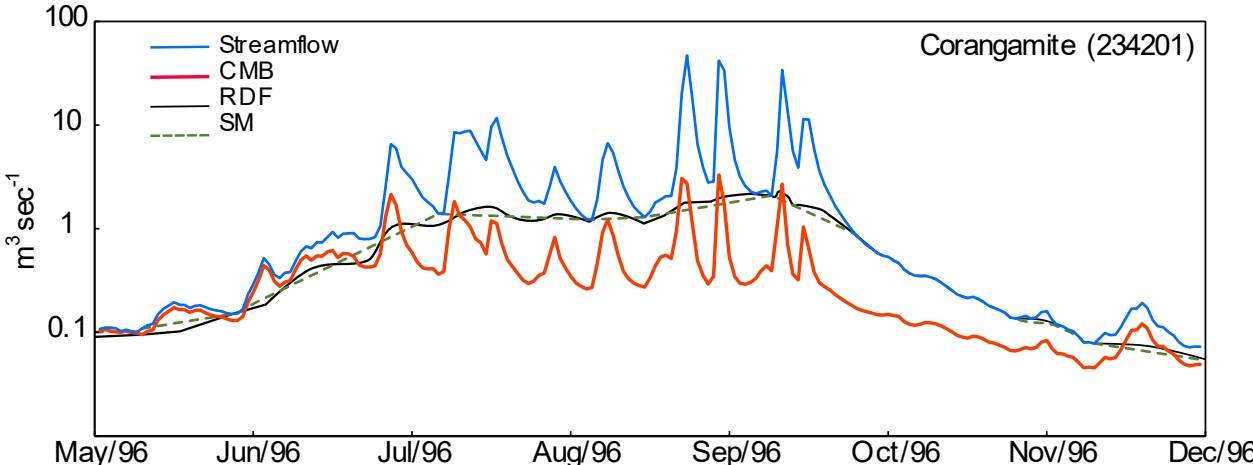

**Fig. 6.** Baseflow fluxes estimated from the CMB and calibrated RDF and SM methods for a six-month period from gauging station 234201 in the Corangamite catchment.

### 5.3. Calibration of baseflow methods

Because long-term $SC_r$ records are less common than streamflow records, the CMB method is sometimes used to calibrate other techniques to extend the estimates of groundwater inflows to times when SC data are not available or to similar catchments (Stewart et al., 2007; Gonzales et al., 2009; Saraiva Okello et al., 2018; Hagedorn, 2020). Yang et al. (2021) highlighted that different biases in the RDF and CMB techniques introduced problems for cross calibration. However, from a pragmatic viewpoint, such calibrations would be valuable if they were possible. In the catchments studied here, calibrating the RDF and SM techniques using the long-term BFI estimates from the CMB proved unsuccessful in estimating annual baseflow. This again probably results from some of the intermediate stores of water (especially bank storage) contributing to streamflow over months to years. Additionally, yearly to decadal variations in groundwater recharge may result in long-term variations in the proportion of groundwater in the baseflow component and its salinity. This is certainly the case in southeast Australia where groundwater heads have not yet recovered following the

prolonged droughts between 1996 and 2010 (Chen et al., 2016; Department of Environment, Land, Water and Planning, 2021). In this study, the constant $SC_b$ approach yields reasonable correlations between annual BFI and annual streamflow in some catchments (Figs 4, S5-S8; Table 1) that may be used to extend the baseflow estimates to periods when SC data are not available.

**6. Conclusions**

This study demonstrates some of the complexities in estimating baseflow, which has implications for assessing inflows of contaminated groundwater, understanding catchment water balances, and determining the impacts of near-river groundwater extraction on streams. In particular, these and similar rivers may only be entirely fed by groundwater in dry years. Assuming that the rivers are sustained by groundwater at low flows each year or

355 analysing only a few years of data from high rainfall periods may result in groundwater inflows being overestimated using the CMB method. SC records of several years to decades that include both low- and high-flow years are ideally required to evaluate and apply the CMB method. In addition, SC values from near-river groundwater could be used to assess whether and when the rivers are entirely sustained by groundwater inflows. The long-term variability and spatial heterogeneity of groundwater inflows also severely complicates efforts to

360 calibrate hydrograph-based techniques using the CMB method.

The high SC values of groundwater in southeast Australia result in the streams having high and variable SC values. Although only a subset of streams was analysed in this study, SC values of other rivers in this region are almost invariably highest during low flow periods in drier years (Department of Land, Water and Planning 2021), implying similar behaviour. Other semi-arid to temperate catchments globally are likely to behave in a

365 similar manner, although this may not be as obvious where the groundwater SC values are lower. Higher rainfall catchments may never be totally sustained by groundwater inflows as interflow and bank storage waters may always be present (McCallum et al., 2010; Cranswick and Cook, 2015; Rhodes et al., 2017; Cartwright and Irvine, 2020). In those cases, the CMB method may still be able to estimate the relative importance of groundwater contributions to baseflow, especially if independent estimates of groundwater SC can be made.

The CMB calculations also imply that groundwater input to rivers is irregular over short timescales (days to months); however, whether baseflow as whole varies smoothly is not clear. Understanding whether that is the case is important as separation techniques based on streamflow data assume a relatively smooth variation in baseflow. There is an increasing number of analytes that may be measured autonomously over extended periods and detailed multi component geochemical studies that can separate different sources of water may help resolve

this question. For example, nitrate is commonly elevated in near-surface waters compared with regional groundwater (e.g. Duan et al., 2014; Bowes et al., 2015) and stable isotopes (Klaus and McDonnell, 2013; Tweed et al., 2016) can also be used to separate different water sources. The physical response of streamflow to rainfall (celerity) is commonly decoupled from the flow of water and solutes stored within catchments (McDonnell and Beven, 2014). It may be more realistic to conceive of the smooth baseflow variations as the

physical response of the catchment rather than reflecting the input of water from specific stores.

## 7. Data Availability

All data is freely available from the Department of Environment, Land, Water and Planning Water Measurement site https://data.water.vic.gov.au/

## 8. Competing Interests

The author declares that he has no conflicts of interest.

## 9. Acknowledgements

Comments from two anonymous reviewers helped clarify several aspects of this paper.

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

**Tables**

**Table 1.** Summary of baseflow estimates from southeast Australia.

| Station[a] | Area | %Flow | Max SC[b] | BFI[c] | BFI[c] | $R^{2\,d}$ | $R^{2\,d}$ | $R^{2\,e}$ | $R^{2\,e}$ | $R^{2\,e}$ | $R^{2\,e}$ |
|---|---|---|---|---|---|---|---|---|---|---|---|
| | | | µS cm$^{-1}$ | CMBc | CMBv | Q-CMBc | Q-CMBv | CMBc-SM | CMBv-SM | CMBc-RDF | CMBv-RDF |
| Barwon Catchment | | | | | | | | | | | |
| 233200 | 2713 | 100 | 3309 | 0.34[f] | 0.42 | **0.70** | *0.67* | *0.55* | *0.57* | 0.14 | 0.31 |
| 233201 | 1052 | 96 | 4740 | 0.17 | 0.32 | *0.65* | 0.18 | 0.11 | 0.10 | 0.01 | 0.07 |
| 233211 | 88 | 33 | 25762 | 0.11 | 0.15 | **0.76** | **0.78** | 0.13 | 0.03 | 0.00 | 0.05 |
| 233213 | 839 | 99 | 2475 | 0.37 | 0.50 | **0.79** | **0.71** | *0.73* | *0.55* | 0.44 | *0.60* |
| 233218 | 1269 | 95 | 11200 | 0.10 | 0.20 | **0.77** | 0.14 | 0.20 | 0.47 | 0.03 | 0.10 |
| 233223 | 57 | 50 | 7622 | 0.11 | 0.13 | **0.90** | **0.88** | *0.69* | **0.70** | 0.38 | 0.41 |
| 233224 | 593 | 98 | 3494 | 0.17 | 0.34 | *0.66* | 0.44 | *0.52* | 0.29 | 0.20 | 0.25 |
| 233247 | 864 | 97 | 4190 | 0.19 | 0.36 | *0.67* | 0.10 | *0.51* | 0.27 | 0.12 | 0.28 |
| 233250 | 5 | 41 | 12959 | 0.09 | 0.16 | *0.60* | 0.06 | 0.08 | 0.05 | 0.01 | 0.02 |
| | | | | | | | | | | | |
| Corangamite Catchment | | | | | | | | | | | |
| 234200 | 324 | 83 | 4547 | 0.22 | 0.26 | **0.93** | **0.89** | *0.65* | **0.72** | 0.31 | 0.14 |
| 234201 | 1158 | 100 | 11922 | 0.17 | 0.21 | **0.95** | **0.92** | **0.79** | **0.81** | 0.43 | 0.46 |
| 234209 | 45 | 86 | 7372 | 0.16 | 0.30 | **0.78** | *0.61* | 0.35 | 0.32 | 0.09 | 0.21 |
| 234212 | 231 | 99 | 21790 | 0.08 | 0.15 | **0.88** | **0.70** | **0.79** | *0.62* | 0.48 | *0.55* |
| | | | | | | | | | | | |
| Goulburn Catchment | | | | | | | | | | | |
| 405212 | 337 | 77 | 2370 | 0.15 | 0.37 | **0.81** | 0.48 | 0.10 | 0.25 | 0.02 | 0.00 |
| 405226 | 787 | 69 | 498 | 0.30 | 0.42 | 0.30 | 0.29 | 0.40 | 0.25 | 0.21 | 0.28 |
| 405240 | 609 | 70 | 1650 | 0.26 | 0.28 | **0.82** | **0.70** | 0.14 | 0.26 | 0.01 | 0.24 |
| 405246 | 164 | 36 | 690 | 0.21 | 0.43 | 0.53 | 0.31 | 0.04 | 0.01 | 0.00 | 0.00 |
| | | | | | | | | | | | |
| Loddon Catchment | | | | | | | | | | | |
| 407211 | 1850 | 74 | 16862 | 0.05 | 0.11 | **0.82** | **0.72** | **0.87** | **0.84** | 0.00 | 0.09 |
| 407239 | 137 | 56 | 2679 | 0.05 | 0.14 | **0.89** | **0.78** | *0.50* | 0.45 | 0.01 | 0.01 |
| 407252 | 2850 | 99 | 36918 | 0.13 | 0.25 | *0.69* | 0.18 | 0.01 | 0.05 | 0.00 | 0.26 |
| 407284 | 650 | 54 | 28667 | 0.04 | 0.13 | 0.31 | 0.01 | 0.11 | 0.05 | 0.12 | 0.52 |
| 407288 | 124 | 69 | 12507 | 0.13 | 0.18 | **0.86** | 0.41 | 0.23 | 0.08 | 0.29 | 0.01 |
| 407289 | nm | 98 | 1869 | 0.24 | 0.31 | 0.00 | 0.04 | 0.00 | 0.01 | 0.00 | 0.00 |

a: Department of Environment, Land, Water and Planning (2021), Figs S1-S4

b: Maximum SC over the monitoring period

c: BFI from the Constant (CMBc) and Variable (CMBv) CMB methods

d: Correlations between streamflow (Q) and BFI from the Constant (CMBc) and Variable (CMBv) CMB
methods

e: Correlations of BFI between the Constant (CMBc) and Variable (CMBv) CMB methods and the RDF and SM methods calibrated to the BFI from the CMB

f: Normal type $R^2$ <0.5, *italic* type $0.5 \geq R^2$ <0.7, bold type $R^2 \geq 0.7$

**Table 2**. Comparison of annual BFI estimates from the Barwon River.

| Single SC$_b$$^a$ | | | | | | Variable SC$_b$$^a$ | | | | |
|---|---|---|---|---|---|---|---|---|---|---|
| Station | 223200 | 223201 | 223218 | 223244 | 223247 | 223200 | 223201 | 223218 | 223244 | 223247 |
| Year | | | | | | | | | | |
| 1994-1995 | 0.53 | | 0.13 | 0.19 | | 0.64 | | 0.37 | 0.35 | |
| 1995-1996 | 0.26 | | 0.05 | 0.11 | 0.16 | 0.32 | | 0.17 | 0.22 | 0.34 |
| 1996-1997 | 0.24 | | 0.07 | 0.11 | 0.16 | 0.32 | | 0.32 | 0.32 | 0.40 |
| 1997-1998 | 0.54 | | 0.14 | 0.25 | 0.22 | 0.68 | | 0.35 | 0.56 | 0.47 |
| 1998-1999 | 0.43 | | 0.10 | 0.16 | | 0.48 | 0.71 | 0.20 | 0.35 | |
| 1999-2000 | 0.58 | 0.25 | 0.19 | 0.28 | 0.26 | 0.61 | 0.31 | 0.32 | 0.62 | 0.53 |
| 2000-2001 | 0.31 | 0.16 | 0.09 | 0.19 | 0.16 | 0.36 | 0.28 | 0.13 | 0.34 | 0.28 |
| 2001-2002 | 0.33 | 0.17 | 0.08 | 0.17 | 0.18 | 0.40 | 0.38 | 0.17 | 0.31 | 0.29 |
| 2002-2003 | 0.49 | 0.22 | 0.13 | 0.23 | 0.25 | 0.54 | 0.36 | 0.27 | 0.44 | 0.48 |
| 2003-2004 | 0.38 | 0.20 | 0.10 | 0.23 | 0.21 | 0.41 | 0.38 | 0.21 | 0.43 | 0.45 |
| 2004-2005 | 0.34 | 0.16 | 0.09 | 0.16 | 0.18 | 0.42 | 0.55 | 0.24 | 0.34 | 0.46 |
| 2005-2006 | 0.45 | 0.24 | 0.14 | 0.24 | 0.24 | 0.62 | 0.35 | 0.31 | 0.52 | 0.59 |
| 2006-2007 | 0.58 | 0.21 | 0.14 | 0.22 | 0.23 | 0.69 | 0.17 | 0.17 | 0.39 | 0.32 |
| 2007-2008 | 0.33 | 0.17 | 0.08 | 0.14 | 0.19 | 0.37 | 0.34 | 0.09 | 0.32 | 0.20 |
| 2008-2009 | 0.53 | 0.27 | 0.15 | 0.28 | 0.24 | 0.65 | 0.47 | 0.29 | 0.56 | 0.38 |
| 2009-2010 | 0.41 | 0.23 | 0.12 | 0.24 | 0.24 | 0.54 | 0.23 | 0.26 | 0.44 | 0.45 |
| 2010-2011 | 0.26 | 0.18 | 0.09 | 0.17 | 0.18 | 0.30 | 0.34 | 0.19 | 0.33 | 0.25 |
| 2011-2012 | 0.33 | 0.15 | 0.09 | 0.16 | 0.16 | 0.43 | 0.27 | 0.20 | 0.33 | 0.31 |
| 2012-2013 | 0.28 | 0.16 | 0.07 | 0.16 | 0.18 | 0.38 | 0.33 | 0.12 | 0.35 | 0.29 |
| 2013-2014 | 0.31 | 0.16 | 0.08 | 0.18 | 0.17 | 0.39 | 0.41 | 0.13 | 0.39 | 0.37 |
| 2014-2015 | 0.39 | 0.20 | 0.12 | 0.23 | 0.23 | 0.51 | 0.33 | 0.25 | 0.36 | 0.45 |
| 2015-2016 | 0.51 | 0.26 | 0.19 | 0.31 | 0.26 | 0.59 | 0.18 | 0.25 | 0.35 | 0.38 |
| 2016-2017 | 0.39 | 0.18 | 0.08 | 0.18 | | 0.49 | 0.23 | 0.10 | 0.23 | |
| 2017-2018 | 0.38 | 0.17 | 0.10 | 0.20 | | 0.48 | 0.32 | 0.13 | 0.30 | |
| 2018-1019 | 0.38 | 0.20 | 0.11 | 0.23 | 0.24 | 0.45 | 0.56 | 0.15 | 0.35 | 0.40 |
| 2019-2020 | 0.33 | 0.17 | 0.11 | 0.23 | 0.25 | 0.46 | | 0.24 | 0.50 | 0.55 |
| R$^2$$^b$ | | **0.70** | **0.74** | *0.57* | *0.63* | | | 0.05 | 0.33 | 0.38 | 0.27 |

a: CMB calculations using the single and variable SC values of baseflow

b: Correlation of annual BFI with station 223200; normal type $R^2$ <0.5, *italic* type $0.5 \geq R^2$ <0.7, **bold** type $R^2$ ≥0.7

