# Peer review of "Implications of variations in stream specific conductivity for estimating baseflow using chemical mass balance and calibrated hydrograph techniques"

_Hydrology and Earth System Sciences, 2021_

## Author Comment (AC1)

Response to Reviewer #1

The reviewer is thanked for their comments on the manuscript. The responses and proposed modifications are outlined below (in blue)

**General comments:**

This manuscript discusses in detail the influence of the fluctuation of the streamflow conductivity of the four basins in southern Australia on the separation results of the conductivity mass balance method, and also discusses the influence of the correction effect of the filtering method and the sliding minima method. In my opinion, the content of this manuscript is meaningful. It will help researchers analyze the differences between different baseflow separation results, and can guide the correction between different separation methods. I think this manuscript can be published after appropriate revisions. Since my mother tongue is not English, I did not comment on grammar, etc. Below are some of my suggestions.

These general comments indicate that the main message of the paper has been conveyed

 **Specific comments:**

1) Line 32. "that that" may be repeated.

Yes, there should only be one "that". This will be corrected.

2) Lines 37-39. "Some of these components … much older." I think the meaning of this sentence may be inaccurate. It should be the infiltration of recent or ancient rainfall.

This sentence was awkward. What it was trying to convey was that rivers are fed by waters from within the catchment that have a range of residence times. It will be reworded along the lines of: "Some of the components of baseflow (e.g., bank return waters and interflow) have short residence times represent whereas others, notably regional groundwater, are generally much older."

3) Line 141. "$SC_b$. is based on the SC of the river during low flows using two methods for estimating $SC_b$ were used." This sentence is confusing, please modify it.

Yes this is a poorly worded sentence. It will be reworded as: "$SC_b$. is based on the SC of the river during low flows; two methods for estimating $SC_b$ were used".

4) Lines 183-184. It is feasible to adopt the recommended value of recession coefficients. However, the recession coefficients of different watersheds are likely to have certain differences, and it can be easily determined through recession analysis. So I suggest you determine it through recession analysis.

Agreed. The recession constants were calculated for the individual catchments and can be used for the calculations. It makes little difference to the numerical results and no difference to the overall conclusions. The range of values was 0.92 to 0.95 with a median of 0.93 (which is what was used for the calculations in the paper).

5) Line 195, Figure 2. Lack of legend for baseflow conductivity.

Agreed. Legend will be amended

6) Line 230, Figure 4. The legend for points and lines is missing.

Agreed. Legend will be added

7) Line 247, Figure 5. The legend for the dots in different colors is missing.

Agreed. Legend will be amended

---

## Author Comment (AC2)

Response to Reviewer #2

The reviewer is thanked for their comments on the manuscript. The responses and proposed modifications are outlined below (in blue)

If I understand this piece of science correctly, it is a valuable criticism of the CMB method during baseflow estimation. This might be (very) interesting for the community as many studies before criticize (pure) hydrograph separation for its missing physical justification, i.e. call for tracer or isotope supported baseflow estimates. Here, the study argues that CMB calculations during specific flow periods might not be applicable to the entire flow series to gain a (valuable) baseflow estimate.

Yes, that was the main message of the paper and hopefully it will be valuable as suggested.

I guess the paper could be even stronger if more detailed information was given how a valuable CMB/SC method should look like (i.e., what kind of flow periods should at least be considered to reduce bias in baseflow estimation, see below).

This can be done. The Conclusions (lines 309-319) contain some of that information but it can be made more explicit. If one wants to estimate groundwater inflows, then there needs to be a strategy that allows the SC of groundwater to be constrained. This can include ensuring that SC records exist over low flow drought periods when the intermediate sources of water have largely disappeared. SC values of groundwater from near the river would also be useful.

A statement along the lines of "To better estimate groundwater inflows using the CMB technique, the SC records ideally need to extend over drought periods with low streamflow when groundwater dominates the baseflow component. SC values from near-river groundwater from within the catchment may also be useful in assessing this" will be added.

I am not sure if the presented bias in baseflow estimation during high SC periods can be transferred 1:1 to other regions than Australia. Or in other words, are the found deficiencies of the presented methods also an issue in more humid catchments, i.e., other regions of the world where typical ranges of SC might be very different to those measured in this study?

That is an interesting and important question. It is likely that similar constraints apply in many catchments. The Barwon catchment is temperate (rainfall of 600-1050 mm) and is probably the one that best illustrates the points in the paper. The key issue is the salinity of the groundwater that in these catchments (and in others) is high by global standards, and which makes the contrasts between years more obvious. It may be that higher rainfall catchments are never only fed by groundwater (i.e. that there is always some component of interflow or bank return waters in the baseflow component).

This again can be explored in the Conclusions, something along the lines of: "Because the groundwater in these southeast Australian catchments has high salinities, the differences in river SC values during dryer and wetter years is obvious. Other semi-arid to temperate catchments globally are likely to behave in a similar manner, although the differences may not be as marked if the groundwater SC values are lower. Catchments with higher rainfall may never be totally sustained by groundwater inflow; however, the CMB method may still be useable to estimate the relative importance of groundwater contributions to baseflow, especially if independent estimates of groundwater SC can be made".

A further concern in this perspective is the selection of catchments that are used to justify the outcomes of the study. I am not sure if the reference to the Supplement is enough to understand the characteristics of the study catchments (as there is also no map or other topographic or hydrogeological information on this catchments). At this point I ask myself how much regional distinctions are in the study and what about the transferability of the results (see above). To judge this, the reader might need more details on the catchments what from my point of view can be easily done by transferring information from supplement to the paper.

More information can be made available. However, this would be better kept in the Supplement as the paper is focussed more on the methodology rather than the case studies. This is similar to the approach followed by the HESS papers of Yang et al. (2021) and Stoelze et al. (2020) (both cited in the paper), which also focussed on the methodology. As to transferability, several other streams in southeast Australia where we have good streamflow and SC records show similar behaviour (as noted in the answer to the previous question, this region is ideal for this analysis due to the high salinity of the groundwater). The catchments chosen were done so as they contained several suitable streams for the analysis (no reservoirs, little groundwater use). There is little value in adding streams from more catchments, although a comment in the Discussion saying that "Streams elsewhere in southeast Australia show similar SC variations between low- and high-flow years (Department of Environment, Land, Water and Planning, 2021), implying that the variation in baseflow components identified here are likely to be common" would help.

More detail could be added to the Supplement, specifically catchment maps, topography, climate and landuse. However, to avoid distractions from the methodological aspects of the paper, it would be preferable to keep most of these details in the Supplement with a few more details at the start of Section 2.1.

The study proposes a multi geochemical analysis in larger rivers to identify many/more sources of water: It would be nice to be more concrete here, e.g., what kind of geochemical analysis are needed, during which seasons or flow periods and what is meant with larger catchments. I doubt that larger catchments will offer a clearer signal as with increasing catchment area also often human interactions increase and regional groundwater systems will become more important. However, it might be worth to gain an additional review for this interesting study from the isotope/tracer or hydrogeological community.

This section can be filled out, although I agree with the reviewer about the challenges. Basically, a tracer such as nitrate (concentrated in near-surface waters and can be measured autonomously) would be most useful. An explanation such as: "There are an increasing number of tracers that can be measured autonomously that may prove useful in this regard. For example, nitrate which is generally concentrated in surface runoff, soil water, and interflow but may be depleted in deeper groundwater, may help distinguish between water sources contributing to baseflow. Stable isotopes, which may also be measured continuously, may also be useable to differentiate water sources".

Minor comments

Fig. 5: What is the difference between the blue and white points (here circles and squares)?

They relate to the two different methods of estimating $SC_b$. The legend and the caption will be amended to explain that

Fig.3: A lot of overplotting is going on here. A density scatterplot might help out to see more details of the point clouds.

There are a lot of data on the plots, but it is important to show all the points. A density scatterplot is less useful as it is the Q vs SC trend that is important. Reducing the symbol size and modifying the colours will help the plots look a little clearer,

Is the filter parameter of 0.93 justified by other studies in the same region or is it just a value from literature? Normally it is recommended to have values between 0.95 and 0.90 and the specific values has a high impact on the actual baseflow estimate.

The 0.93 was the median value of the measured values for the catchments. The actual values ranged from 0.92 to 0.95. As noted in the response to Reviewer #1, the recession constants were calculated for the individual catchments and ranged from 0.92 to 0.95. The individual values can be used for the calculations in the revised version but it makes little difference to the numerical results and no difference to the overall conclusions.

In general, the axes labels of most figures are too small.

These will be enlarged

The SM method is based on variable N. Is N somewhere reported for the specific catchments? And, is the assumption of N being a function of catchment area really valuable?

N values can be added to the text and Table 1. The paper did not assume that N vas a simple function of catchment area (I agree that that is probably an oversimplification). As outlined on lines 173-177, N was estimated so that the long-term baseflow method estimated from the sliding minima method was the same as that from the CMB (as proposed by Stewart et al., 2007). The values of N calculated in this way are larger than those based on area (which one would expect if the hydrograph estimated total baseflow and the CMB estimated mainly the groundwater input). Stoelzle et al. (2020) used a different approach to estimating N (based on the breakpoints in the BFI vs. N trends), but one that also produced different N values to those based on catchment area. The statement on lines 173-177 can be made clearer.

---

## Author Response (AR1)

The comments of both reviewers were addressed as outlined below (in blue). The figures were also modified to make the lines and symbols more distinct for readers with colour deficient vision and to ensure consistency of fonts. Line numbers refer to the marked version.

**Reviewer #1:**

1) Line 32. "that that" may be repeated.

Changed to "that" (line 32)

2) Lines 37-39. "Some of these components … much older." I think the meaning of this sentence may be inaccurate. It should be the infiltration of recent or ancient rainfall.

This sentence was removed as the age of the water was not referred to in the paper. The previous sentences (lines 33-37): "Baseflow represents water stored in the catchment that sustains streamflow between precipitation events. Regional groundwater may be a significant component of baseflow in gaining rivers; however, displaced soil water, interflow, bank return flows, snow melt, and/or water stored in floodplain pools can also be important" contain the important information as to the different water stores.

3) Line 141. "$SC_b$. is based on the SC of the river during low flows using two methods for estimating $SC_b$ were used." This sentence is confusing, please modify it.

The description of the calculation of $SC_b$ has been reworded (lines 189-192). "In common with other studies (Sanford et al., 2011; Miller et al., 2014, 2015, 2016; Cartwright et al., 2014; Rumsey et al., 2015), $SC_b$ is estimated from the SC of the river during low flows. Two methods for estimating $SC_b$ were used. The *Variable SC* approach estimates daily $SC_b$ values by interpolating between high $SC_r$ values in successive water years, which assumes that the river is entirely fed by groundwater each year during low flows (as in Fig. 1)".

4) Lines 183-184. It is feasible to adopt the recommended value of recession coefficients. However, the recession coefficients of different watersheds are likely to have certain differences, and it can be easily determined through recession analysis. So I suggest you determine it through recession analysis.

The recession constants were recalculated for individual hydrographs. This is reported on lines 234-237: "In Eq. (2), a is the recession constant which was estimated from the falling limbs of the hydrograph following Nathan and McMahon (1990) and Eckhardt (2005). a varies between 0.92 and 0.95 with a median value of 0.93 (Table S1)." Table S1 in the Supplement summarises the parameters used in the filters and sliding minimum analyses. The changes to a made no substantial difference to the calculations.

5) Line 195, Figure 2. Lack of legend for baseflow conductivity.

The legend was corrected (line 249).

6) Line 230, Figure 4. The legend for points and lines is missing.

The legend was added (line 285).

7) Line 247, Figure 5. The legend for the dots in different colors is missing.

The meaning of the closed and open symbols was added to the caption (lines 305-308).

**Reviwer #2**

Some of the general comments made by Reviewer 2 were addressed in the original version, but were scattered through the different sections of the Discussion. In the revised version, the more general themes have been moved to the Conclusions with additional details.

I guess the paper could be even stronger if more detailed information was given how a valuable CMB/SC method should look like (i.e., what kind of flow periods should at least be considered to reduce bias in baseflow estimation, see below).

This has been made more explicit in the Conclusions (lines 374-377): "SC records of several years to decades that include both low- and high-flow years are ideally required to evaluate and apply the CMB method. In addition, SC values from near-river groundwater could be used to assess whether and when the rivers are entirely sustained by groundwater inflows."

I am not sure if the presented bias in baseflow estimation during high SC periods can be transferred 1:1 to other regions than Australia. Or in other words, are the found deficiencies of the presented methods also an issue in more humid catchments, i.e., other regions of the world where typical ranges of SC might be very different to those measured in this study?

This is also now addressed in the Conclusions (lines 380-388): "The high SC values of groundwater in southeast Australia result in the streams having high and variable SC values. Although only a subset of streams was analysed in this study, SC values in the rivers in this region are almost invariably highest during low flow periods in drier years (Department of Land, Water and Planning 2021), implying similar behaviour. Other semi-arid to temperate catchments globally are likely to behave in a similar manner, although this may not be as obvious where the groundwater SC values are lower. Higher rainfall catchments may never be totally sustained by groundwater inflows as interflow and bank storage waters may always be present (McCallum et al., 2010; Cranswick and Cook, 2015; Rhodes et al., 2017; Cartwright and Irvine, 2020). In those cases, the CMB method may still be able to estimate the relative importance of groundwater contributions to baseflow, especially if independent estimates of groundwater SC can be made."

A further concern in this perspective is the selection of catchments that are used to justify the outcomes of the study. I am not sure if the reference to the Supplement is enough to understand the characteristics of the study catchments (as there is also no map or other topographic or hydrogeological information on this catchments). At this point I ask myself how much regional distinctions are in the study and what about the transferability of the results (see above). To judge this, the reader might need more details on the catchments what from my point of view can be easily done by transferring information from supplement to the paper.

In response to this (and also the request from the Associate Editor), the descriptive material from the Supplement was moved to the main body of the paper (Section 2, lines 113-157). The start of Section 3 (lines 160-166) was reworded to avoid repetition. Catchment maps showing the location of the gauging stations and landuse were added to the Supplement (Figs S1-S4).

The study proposes a multi geochemical analysis in larger rivers to identify many/more sources of water: It would be nice to be more concrete here, e.g., what kind of geochemical analysis are needed, during which seasons or flow periods and what is meant with larger catchments. I doubt that larger catchments will offer a clearer signal as with increasing catchment area also often human interactions

increase and regional groundwater systems will become more important. However, it might be worth to gain an additional review for this interesting study from the isotope/tracer or hydrogeological community.

This has also been addressed in the Conclusions (lines 392-396): "There is an increasing number of analytes that may be measured autonomously over extended periods and detailed multi component geochemical studies can separate different sources of water may help resolve this question. For example, nitrate is commonly elevated in near-surface waters compared with regional groundwater (e.g. Duan et al., 2014; Bowes et al., 2015) and stable isotopes (Klaus and McDonnell, 2013; Tweed et al., 2016) can also be used to separate different water sources."

Minor comments

Fig. 5: What is the difference between the blue and white points (here circles and squares)?

The figure caption has been amended to explain the closed and open symbols (lines 305-308).

Fig.3: A lot of overplotting is going on here. A density scatterplot might help out to see more details of the point clouds.

The symbols on these plots have been changed to make them clearer (line 261).

Is the filter parameter of 0.93 justified by other studies in the same region or is it just a value from literature? Normally it is recommended to have values between 0.95 and 0.90 and the specific values has a high impact on the actual baseflow estimate.

As discussed above, the recession constants are now calculated for individual hydrographs. This is reported on lines 234-237: "In Eq. (2), a is the recession constant which was estimated from the falling limbs of the hydrograph following Nathan and McMahon (1990) and Eckhardt (2005). a varies between 0.92 and 0.95 with a median value of 0.93 (Table S1)." Table S1 in the Supplement summarises the parameters used in the filters and sliding minimum analyses.

In general, the axes labels of most figures are too small.

The text size on the figures has been increased

The SM method is based on variable N. Is N somewhere reported for the specific catchments? And, is the assumption of N being a function of catchment area really valuable?

More detail has been provided and the values of N, a, and $BFI_{max}$ have been included in Table S1. Lines 292-296 state: "Adjusting the SM technique by varying N in Eq. (2) and the RDF by varying $BFI_{max}$ in Eq. (3) allows the total BFI estimates from these methods to be brought into agreement with those calculated using the CMB (Tables 1 and S1). For the SM technique, N values are as high as 35 days and are higher in catchments with low BFI values (Table S1). Values of $BFI_{max}$ are inversely proportional to the calculated BFI, and are as low as 0.07 (Table S1)."